# SITUATING SENTENCE EMBEDDERS WITH NEAREST NEIGHBOR OVERLAP

## ABSTRACT

As distributed approaches to natural language semantics have developed and diversified, embedders for linguistic units larger than words (e.g., sentences) have come to play an increasingly important role. To date, such embedders have been evaluated using benchmark tasks (e.g., GLUE) and linguistic probes. We propose a comparative approach, *nearest neighbor overlap* (N2O), that quantifies similarity between embedders in a task-agnostic manner. N2O requires only a collection of examples and is simple to understand: two embedders are more similar if, for the same set of inputs, there is greater overlap between the inputs' nearest neighbors. We use N2O to compare 21 sentence embedders and show the effects of different design choices and architectures.

## 1 INTRODUCTION

Continuous embeddings—of words and of larger linguistic units—are now ubiquitious in NLP. The success of self-supervised pretraining methods that deliver embeddings from raw corpora has led to a proliferation of embedding methods, with an eye toward "universality" across NLP tasks. Our focus here is on **sentence embedders**, and specifically their evaluation. As with most NLP components, intrinsic (e.g., Conneau et al., 2018) and extrinsic (e.g., GLUE; Wang et al., 2019) evaluations have emerged for sentence embedders.

Our approach, **nearest neighbor overlap** (N2O), is different: it compares a pair of embedders in a linguistics- and task-agnostic manner, using only a large unannotated corpus. The central idea is that two embedders are more similar if, for a fixed query sentence, they tend to find nearest neighbor sets that overlap to a large degree. By drawing a random sample of queries from the corpus itself, N2O can be computed on in-domain data without additional annotation, and therefore can help inform embedder choices in applications such as text clustering (Cutting et al., 1992), information retrieval (Salton & Buckley, 1988), and open-domain question answering (Seo et al., 2018), among others.

After motivating and explaining the N2O method (§2), we apply it to 21 sentence embedders (§3-4). Our findings (§5) reveal relatively high functional similarity among averaged static (noncontextual) word type embeddings, a strong effect of the use of subword information, and that BERT and GPT are distant outliers. In §6, we demonstrate the robustness of N2O across different query samples and probe sizes. We also illustrate additional analyses made possible by N2O: identifying embedding-space neighbors of a query sentence that are stable across embedders, and those that are not (§7); and probing the abilities of embedders to find a known paraphrase (§8). The latter reveals considerable variance across embedders' ability to identify semantically similar sentences from a broader corpus.

## 2 CORPUS-BASED EMBEDDING COMPARISON

We first motivate and introduce our nearest neighbor overlap (N2O) procedure for comparing embedders (maps from objects to vectors). Although we experiment with sentence embedders in this paper, we note that this comparison procedure can be applied to other types of embedders (e.g., phrase-level or document-level).[1]

---

[1] We also note that nearest neighbor search has been frequently used on *word* embeddings (e.g., word analogy tasks).

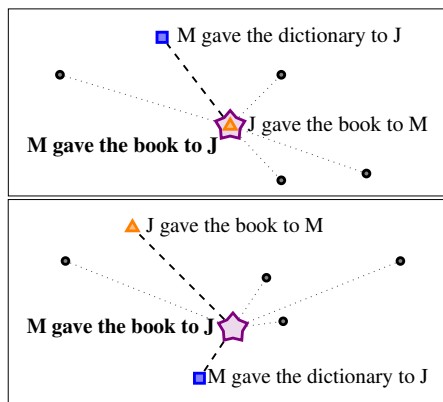

Figure 1: A toy example of two sentence embedders and how they might affect nearest neighbor sentences.

```
function N2O(e_A, e_B, C, k)
   for each query q_j ∈ {q_i}_{i=1}^n do
      neighbors_A ← nearest(e_A, q_j, C, k)
      neighbors_B ← nearest(e_B, q_j, C, k)
      o[j] ← |neighbors_A ∩ neighbors_B|
   end for
   return ∑_j o[j]/(k × n)
end function
```

Figure 2: Computation of N2O for two embedders, $\mathbf{e}_A$ and $\mathbf{e}_B$, using a corpus $C$; the number of nearest neighbors is given by $k$. $n$ is the number of queries $(\mathbf{q}_1 \ldots \mathbf{q}_n)$, which are sampled uniformly from the corpus without replacement. The output is in $[0, 1]$, where 0 indicates no overlap between nearest neighbors for all queries, and 1 indicates perfect overlap.

## 2.1 DESIDERATA

We would like to quantify the extent to which sentence embedders vary in their treatment of "similarity." For example, given the sentence *Mary gave the book to John*, embedders based on bag-of-words will treat *John gave the book to Mary* as being maximally similar to the first sentence, whereas other embedders may treat *Mary gave the dictionary to John* as more similar; our comparison should reflect this intuition. We would also like to focus on using naturally-occuring text for our comparison. Although there is merit in expert-constructed examples (see linguistic probing tasks referenced in §9), we have little understanding of how these models will generalize to text from real documents; many application settings involve computing similarity across texts in a corpus. Finally, we would like our evaluation to be task-agnostic, since we expect embeddings learned from large unannotated corpora in a self-supervised (and task-agnostic) manner to continue to play an important role in NLP.

As a result, we base our comparison on *nearest neighbors*: first, because similarity is often assumed to correspond to nearness in embedding space (e.g., Figure 1); second, because nearest neighbor methods are used directly for retrieval and other applications; and finally, because the nearest neighbors of a sentence can be computed for any embedder on any corpus without additional annotation.

## 2.2 ALGORITHM

Suppose we want to compare two sentence embedders, $\mathbf{e}_A(\cdot)$ and $\mathbf{e}_B(\cdot)$, where each embedding method takes as input a natural language sentence $\mathbf{s}$ and outputs a $d$-dimensional vector. For our purposes, we consider variants trained on different data or using different hyperparameters, even with the same parameter estimation procedure, to be different sentence embedders.

Take a corpus $C$, which is likely to have some semantic overlap in its sentences, and segment it into sentences $\mathbf{s}_1, \ldots, \mathbf{s}_{|C|}$. Randomly select a small subset of the sentences in $C$ as "queries" $(\mathbf{q}_1, \ldots, \mathbf{q}_n)$. To see how similar $\mathbf{e}_A$ and $\mathbf{e}_B$ are, we compute the overlap in nearest neighbor sentences, averaged across multiple queries; the algorithm is in Figure 2. $nearest(\mathbf{e}_i, \mathbf{q}_j, C, k)$ returns the $k$ nearest neighbor sentences in corpus $C$ to the query sentence $\mathbf{q}_j$, where all sentences are embedded with $\mathbf{e}_i$.[2] There are different ways to define nearness and distance in embedding spaces (e.g., using cosine similarity or Euclidean distance); in this paper we use cosine similarity.

We can think about this procedure as randomly probing the sentence vector space (through the $n$ query sentences) from the larger space of the embedded corpus, under a sentence embedder $\mathbf{e}_i$; in some sense, $k$ controls the depth of the probe. The N2O procedure then compares the sets of sentences recovered by the probes.

---

[2]One of these will be the query sentence itself, since we sampled it from the corpus; we assume *nearest* ignores it when computing the $k$-nearest-neighbor lists.

## 3    SENTENCE EMBEDDING METHODS

In the previous section, we noted that we consider a "sentence embedder" to encompass how it was trained, which data it was trained on, and any other hyperparameters involved in its creation. In this section, we first review the broader methods behind these embedders, turning to implementation decisions in §4.

### 3.1    TF-IDF

We consider **tf-idf**, which has been clasically used in information retrieval settings. The tf-idf of a word token is based off two statistics: term frequency (how often a term appears in a document) and inverse document frequency (how rare the term is across all documents). The vector representation of the document is the idf-scaled term frequencies of its words; in this work we treat each sentence as a "document" and the vocabulary-length tf-idf vector as its embedding.

### 3.2    WORD EMBEDDINGS

Because sentence embeddings are often built from word embeddings (through initialization when training or other composition functions), we briefly review notable word embedding methods.

**Static embeddings.**    We define "static embeddings" to be fixed representations of every word type in the vocabulary, regardless of its context. We consider three popular methods: **word2vec** (Mikolov et al., 2013) embeddings optimized to be predictive of a word given its context (continuous bag of words) or vice versa (skipgram); **GloVe** (Pennington et al., 2014) embeddings learned based on global cooccurrence counts; and **FastText** (Conneau et al., 2017), an extension of word2vec which includes character $n$-grams (for computing representations of out-of-vocabulary words).

**Contextual embeddings.**    Contextual word embeddings, where a word token's representation is dependent on its context, have become popular due to improvements over state-of-the-art on a wide variety of tasks. We consider:

- **ELMo** (Peters et al., 2018) embeddings are generated from a multi-layer, bidirectional recurrent language model that incorporates character-level information.

- **GPT** (Radford et al., 2018) embeddings are generated from a unidirectional language model with multi-layer transformer decoder; subword information is included via byte-pair encoding (BPE; Sennrich et al., 2016).

- **BERT** (Devlin et al., 2019) embeddings are generated from a transformer-based model trained to predict (a) a word given both left and right context, and (b) whether a sentence is the "next sentence" given a previous sentence. Subword information is incorporated using the WordPiece model (Schuster & Nakajima, 2012).

**Composition of word embeddings.**    The simplest way to obtain a sentence's embedding from its sequence of words is to average the word embeddings.[3] Despite the fact that averaging discards word order, it performs surprisingly well on sentence similarity, NLI, and other downstream tasks (Wieting et al., 2016; Arora et al., 2017).[4]

In the case of contextual embeddings, there may be other conventions for obtaining the sentence embedding, such as using the embedding for a special token or position in the sequence. With BERT, the `[CLS]` token representation (normally used as input for classification) is also sometimes used as a sentence representation; similarly, the last token's representation may be used for GPT.

---

[3]In the case of GPT and BERT, which yield subword embeddings, we treat those as we would standard word embeddings.

[4]Arora et al. (2017) also suggest including a PCA-based projection with word embedding averaging to further improve downstream performance. However, because our focus is on behavior of the embeddings themselves, we do not apply this projection here.

| Embed. method | Composition | Dim. | Model/data description |
|---|---|---|---|
| tf-idf | n/a | $|V|$ | tf-idf statistics obtained on Gigaword corpus (2010 slice) |
| word2vec | average | 300 | Google News (3B tokens) |
| GloVe | average | 100 | Wikipedia 2014 + Gigaword 5 (6B tokens, uncased) |
| | | 300 | Wikipedia 2014 + Gigaword 5 (6B tokens, uncased) |
| | | 300 | Common Crawl (840B tokens, cased) |
| FastText | average | 300 | Wikipedia + UMBC + statmt.org (16B tokens) |
| | | 300 | " + subword information |
| | | 300 | Common Crawl (600B tokens) |
| | | 300 | " + subword information |
| ELMo | average | 256 | pretrained small model (1 Billion Word Benchmark) |
| | | 1024 | pretrained original model (1 Billion Word Benchmark) |
| | | 1024 | pretrained original/5.5B model (Wikipedia/news) |
| BERT | [CLS] | 768 | pretrained cased/base model (Wikipedia + BooksCorpus) |
| | average | 768 | |
| | [CLS] | 768 | " + finetuning on MultiNLI (matched subset) |
| | average | 768 | |
| GPT | last | 512 | pretrained model (110M parameters) trained on BooksCorpus |
| | average | 512 | |
| InferSent | n/a | 4096 | V1 (GloVe-based) model, trained on SNLI |
| USE | n/a | 512 | deep averaging network (DAN) encoder; multitask training |
| | | 512 | transformer encoder; multitask training |

Table 1: Details of the pretrained sentence embedders we test in this paper. For methods which produce word embeddings, "composition" denotes how a single embedding was obtained from the sentence's word embeddings. ELMo embeddings are averaged across the three bi-LSTM layers; BERT and GPT embeddings come from the final hidden layer. All of the models besides tf-idf and the fine-tuned version of BERT are common pretrained versions; further details are in Appendix A.

## 3.3 ENCODERS

A more direct way to obtain sentence embeddings is to learn an encoding function that takes in a sequence of tokens and outputs a single embedding; often this is trained using a relevant supervised task. We consider two encoder-based methods:

- **InferSent** (Conneau et al., 2017): supervised training on the Stanford Natural Language Inference (SNLI; Bowman et al., 2015) dataset; the sentence encoder provides representations for the premise and hypothesis sentences, which are then fed into a clasifier.

- Universal Sentence Encoder (**USE**; Cer et al., 2018): supervised, multi-task training on several semantic tasks (including semantic textual similarity); sentences are encoded either with a deep averaging network or a transformer.

## 4 EXPERIMENTAL DETAILS

Our main experiment is a broad comparison, using N2O, of the embedders discussed above and listed in Table 1. Despite the vast differences in methods, N2O allows us to situate each in terms of its functional similarity to the others.

**N2O computation.** We describe a N2O sample as, for a given random sample of $n$ queries, the computation of $N2O(\mathbf{e}_A, \mathbf{e}_B, C, k)$ for every pair of sentence embedders as described in §2, using cosine similarity to determine nearest neighbors. The results in §5 are with $k$ (the number of sentences retrieved) set to 50, averaged across five samples of $n = 100$ queries. We illustrate the effects of different $k$ and N2O samples in §6.

**Corpus.** For our corpus, we draw from the English Gigaword (Parker et al., 2011), which contains newswire text from seven news sources. For computational feasibility, we use the articles

from 2010, for a total of approximately 8 million unique sentences.[5] We note preprocessing details (segmentation, tokenization) in Appendix A.

**Queries.**   For each N2O sample, we randomly select 100 ledes (opening sentences) from the news articles of our corpus, and use the same ones across all embedders. Because the Gigaword corpus contains text from multiple news sources covering events over the same time period, it is likely that the corpus will contain semantically similar sentences for a given lede. The average query length is 30.7 tokens (s.d. 10.2); an example query is: "Sandra Kiriasis and brakewoman Stephanie Schneider of Germany have won the World Cup bobsled race at Lake Placid."

**Sentence embedders.**   Table 1 details the sentence embedders we use in our experiments. In general, we use popular pretrained versions of the methods described in §3. We also select pretrained variations of the same method (e.g., FastText embeddings trained from different corpora) to permit more controlled comparisons. In a couple of cases, we train/finetune models of our own: for tf-idf, we compute frequency statistics using our corpus, with each sentence as its own "document"; for BERT, we use the Hugging Face implementation with default hyperparameters,[6] and finetune using the matched subset of MultiNLI (Williams et al., 2018) for three epochs (dev. accuracy 84.1%).

We note that additional embedders are easily situated among the ones tested in this paper by first computing nearest neighbors of the same query sentences, and then computing overlap with the nearest neighbors obtained in this paper. To enable this, the code, query sentences, and nearest neighbors per embedder and query will be publicly available.

## 5   RESULTS

In this section, we present the results from the experiment described in §4. Fig. 3 shows N2O between each pair of sentence embedders listed in Table 1; the values range from 0.04 to 0.62. While even the maximum observed value may not seem large, we reiterate that overlap is computed over two draws of $k = 50$ sentences (nearest neighbors) from approximately 8 million sentences, and even an N2O of 0.04 is unlikely from random chance alone.

**Averages of static word embeddings.**   We first observe that there is generally high N2O among this set of embedders in comparison to other categories (Fig. 4, left). Some cases where N2O is high for variations of the same embedder: `glove-6b-100d` and `glove-6b-300d`, which have different dimensionality but are otherwise trained with the same method and corpus (and to a lesser extent `glove-840b-300d`, which retains casing and is trained on a different corpus); `fasttext-cc` and `fasttext-wiki`, which again are trained with the same method, but different corpora.

The use of subword information, unique to `fasttext-cc-sub` and `fasttext-wiki-sub`, has a large effect on N2O; there is a high (0.52) N2O value for these two and much lower N2O with other embedders, including their analogues without subword information. This effect is also illustrated by measuring, for a given embedder, the average token overlap between the query and its neighbors (see Fig. 5 in Appendix B). As we would expect, subword methods find near neighbors with lower token overlap, because they embed surface-similar strings near to each other.

**tf-idf.**   Unsurprisingly, tf-idf has low N2O with other embedders (even those based on static word embeddings). Like the subword case, we can also use token overlap to understand why this is the case: its nearest neighbors have by far the largest token overlap with the query (0.43).

**Averages of ELMo embeddings.**   We test three ELMo pretrained models across different capacities (`elmo-small`, `elmo-orig`) but the same training data, and across different training data but the same model capacity (`elmo-orig`, `elmo-orig-5.5b`). These two embedder pairs have high N2O (0.42 and 0.55 respectively); the mismatched pair, with both different training data and capacities, has slightly lower N2O (0.38).

---

[5]Because many news articles show up multiple times in the corpus, 23% of sentences in the English Gigaword are exact duplicates of one another; we remove these duplicates.

[6]`https://github.com/huggingface/pytorch-transformers`

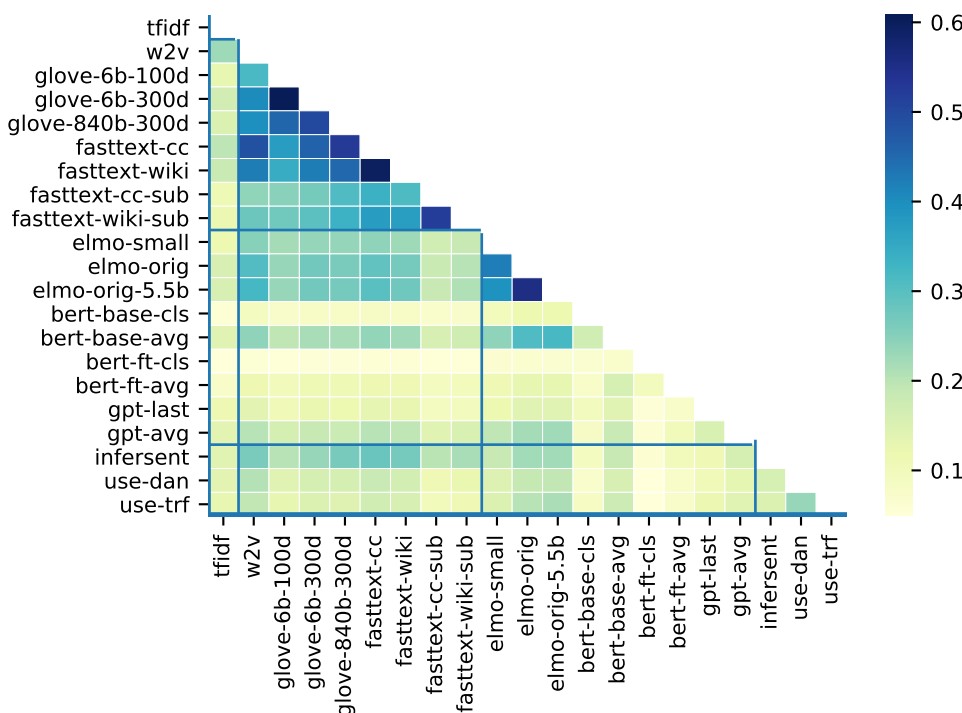

Figure 3: Heatmap of N2O for every pair of sentence embedders in Table 1 for $k = 50$, averaged across five samples of $n = 100$ queries; darker colors indicate higher overlap. A larger version of this plot (annotated with N2O values) is in Appendix B.

**BERT and GPT.** We first find that specific-token representations for BERT or GPT (`bert-base-cls`, `gpt-last`) are outliers compared to other embedders (i.e., low N2O; see Fig. 4). This itself is not unexpected, as the training objectives for both of the pretrained models (without finetuning) are not geared towards semantic similarity the way other embedders are. What is surprising is that this effect seems to hold even for the MultiNLI-finetuned version of BERT (`bert-ft-cls`); if anything, this decreases N2O with other embedders further.[7] Notably, taking *averaged* BERT and GPT embeddings yields higher N2O with other embedders, especially ELMo-based ones. Fig. 6 (Appendix B) plots the N2O values for each embedder compared to all others.

**Encoder-based embedders.** We find that InferSent has highest N2O ($\sim$0.2–0.3) with the embeddings based on averaging, despite InferSent being trained using a NLI task; that said, this is not wholly surprising as the model was initialized using GloVe vectors (`glove-840b-300d`) during training. The USE variants (DAN and Transformer) have fairly distinct nearest neighbors compared to other methods, with highest N2O between each other (0.24).

## 6    ROBUSTNESS AND RUNTIME CONSIDERATIONS

**Varying $k$.** One possible concern is how sensitive our procedure is to $k$ (the number of nearest neighbors from which overlap is computed): we would not want conflicting judgments of how similar two sentence embedders are due to different $k$. To confirm this, we first compute the ranked lists of N2O output for each $k \in \{5, 10, \ldots, 45, 50\}$, where each list consists of all embedder pairs ordered by N2O for that $k$. We then compute Spearman's rank correlation coefficient ($\rho$) between each pair of ranked lists, where 1 indicates perfect positive correlation. We find that the average

---

[7]In preliminary experiments, we also saw similar results with BERT finetuned on the Microsoft Research Paraphrase Corpus (Dolan et al., 2004); that is, the effect does not seem specific to MultiNLI.

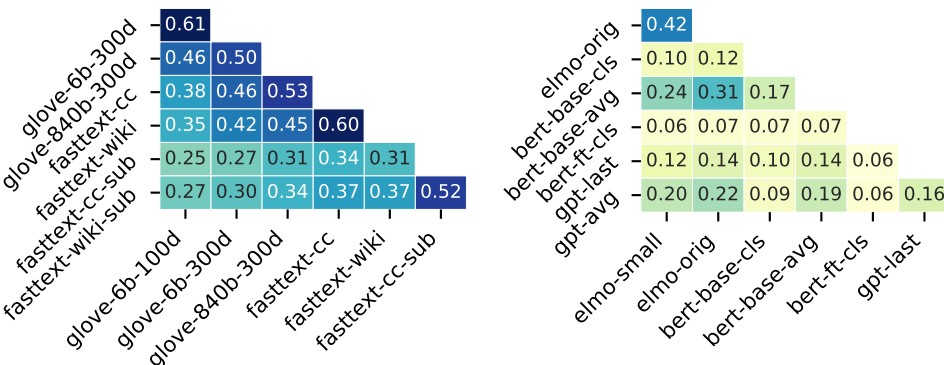

Figure 4: N2O values for a subset of embedders (L: static; R: contextual), $k = 50$.

Spearman's $\rho$ is very high (0.996; min. 0.986) — i.e., the rankings of embedder similarity by N2O are reasonably stable across different values of $k$, even as far as $k = 5$ and $k = 50$.

**Query sampling.** We also examine how the results may vary across different query samples; as noted previously, the presented results are averaged across five samples of $n = 100$ queries each. Standard deviations for N2O values across the five samples range from 0.005 to 0.019 (avg. 0.011). That is, given the range of N2O values being compared, the differences due to different query samples is small. We compute Spearman's $\rho$ across different N2O samples in the same manner as above ($k = 50$) and find an average $\rho$ of 0.994 (min. 0.991).

**Runtime.** A theoretical concern with N2O is that, naively, its computation is linear in the size of the corpus, and to have reasonable semantic overlap within a diverse set of sentences, the corpus should be large. While our implementation of exact nearest neighbor search is sufficiently fast in practice,[8] we provide comments on use of approximate nearest neighbor methods in Appendix C.

## 7 POPULARITY OF NEIGHBORS

In the previous section, we performed a basic comparison between sentence embedders using N2O. Here, we show one kind of analysis enabled by N2O: given a query, which sentences from the corpus $C$ are consistently its neighbors across different embedders? We might expect, for example, that a nearly identical paraphrase of the query will be a "popular" neighbor chosen by most embedders. Table 2 shows an example query with a sentence that is in the 5-nearest neighborhood for all sentence embedders, as well as sentences that are highly ranked for *some* embedder but not in the nearest neighbor sets for *any other* embedder (for larger $k = 50$). Qualitatively, what we find with this example's outlier sentences is that they are often thematically similar in some way (such as fiscal matters in Table 2), but with different participants. We also observe that extremely "popular" neighbors tend to have high lexical overlap with the query.

## 8 QUERY PARAPHRASING

Attempts to derive sentence embeddings that capture semantic similarity are inspired by the phenomenon of paraphrase; in this section, we use nearest neighbors to probe how sentence embedders capture paraphrase. More specifically, we carry out a "needle-in-a-haystack" experiment using the Semantic Textual Similarity Benchmark (STS; Cer et al., 2017). STS contains sentence pairs with human judgments of semantic similarity on a 1–5 continuous scale (least to most similar).

---

[8]Given precomputed sentence embeddings, exact nearest neighbor search across the corpus takes 30s.–1min. (depending on dimensionality) for a batch of $n = 100$ queries and $k = 50$, across two 12-core Intel Xeon CPUs (E5-2960/2.60GHz).

*Query*: Britain's biggest mortgage lender says that average house prices fell 3.6 percent in September, but analysts believe the market isn't that weak.

| Embedder | Rank | Sentence |
|---|---|---|
| all embedders | $\leq 5$ | Average house prices in Britain fell 3.6 percent in September from a month earlier, the country's biggest mortgage lender said Thursday, although analysts believe the market isn't that weak. |
| `bert-base-cls` | 6 | Some analysts say that the December data indicate that consumer spending remains weak, making it harder for the economy to keep a sustained rebound. |
| `bert-ft-avg` | 5 | An industry group says German machinery orders were down 3 percent on the year in January but foreign demand is improving. |
| `gpt-last` | 8 | The economy has since rebound and grew 8.9 percent year-on-year in the second quarter, the central bank said last month, with growth expected to exceed six percent in the full year. |

Table 2: Popular and outlier near neighbors for the given query (top). The first sentence is in the 5-nearest neighborhood for all embedders; the remaining sentences are highly-ranked by the given embedder and outside the 50-nearest neighborhood for all other embedders. See Table 3 (Appendix B) for additional examples.

We take 75 pairs in the 4–5 range from the STS development and test sets where the sentence pair has word-level overlap ratio $< 0.6$ — i.e., near paraphrases with moderately different surface semantics. We also constrain the sentence pairs to come from the newstext-based parts of the dataset. The first sentence in each sentence pair is the "query," and the second sentence is (temporarily) added to our Gigaword corpus. An example sentence pair, scored as 4.6, is: (A) *Arkansas Supreme Court strikes down execution law* and (B) *Arkansas justices strike down death penalty*. We then compute the *rank* of the sentence added to the corpus (i.e., the value of $k$ such that the added sentence is part of the query's nearest neighbors). An embedder that "perfectly" correlates semantic similarity and distance should yield a rank of 1 for the sentence added to the corpus, since that sentence would be nearest to the query.

**Results.** Using mean reciprocal rank (MRR), we find that the larger ELMo models and Infersent do particularly well at placing paraphrase pairs near each other. We also find that averaged BERT and GPT embeddings consistently perform better than the `[CLS]`/final token ones[9]; this is consistent with our earlier observation (§5) that their training objectives may not yield specific-token embeddings that directly encode semantic similarity, hence why they are outliers by N2O. The full table of results is in Table 4 (Appendix B).

## 9 RELATED WORK

Recent comparisons of sentence embedders have been primarily either (1) linguistic probing tasks or (2) downstream evaluations. Linguistic probing tasks test whether embeddings can distinguish surface level properties, like sentence length; syntactic properties, like tree depth; and semantic properties, like coordination inversion. See Ettinger et al. (2016), Adi et al. (2017), Conneau et al. (2018), and Zhu et al. (2018), among others. Downstream evaluations are often classification tasks for which good sentence representations are helpful (e.g., NLI). Evaluations like the RepEval 2017 shared task (Nangia et al., 2017), SentEval toolkit (Conneau & Kiela, 2018), and GLUE benchmark (Wang et al., 2019) seek to standardize comparisons across sentence embedding methods. N2O is complementary to these, providing a task-agnostic way to compare embedders' functionality.

## 10 CONCLUSION

In this paper, we introduce *nearest neighbor overlap* (N2O), a comparative approach to quantifying similarity between sentence embedders. Using N2O, we draw comparisons across 21 embedders. We also provide additional analyses made possible with N2O, from which we find high variation in embedders' treatment of semantic similarity.

---

[9]The BERT results with STS are consistent with concurrent work by Riemers & Gurevych (2019).

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

## A    APPENDIX: IMPLEMENTATION DETAILS

In this section, we include additional implementation details for experiments performed in the paper. Generally, we use parameters consistent with the original work when possible.

**Sentence segmentation.**    We use the `spacy`[10] library (2.0.16) to perform sentence segmentation; for word tokenization, we defer to preferences for the original embedder implementations if specified (see below), or use the `spacy` tokenizer otherwise.

**Tf-idf.**    We use the `gensim` library (3.7.3) implementation of tf-idf,[11] with frequency statistics learned on the 2010 section of the Gigaword corpus (i.e., the same corpus used to find nearest neighbors). For tokenization, we use the Gensim tokenizer and lowercase all word tokens.

**Word2vec.**    We use pretrained 300D Google News embeddings available from Google.[12] We use `spacy` to perform word tokenization and embedding lookup.

---

[10] http://spacy.io
[11] https://radimrehurek.com/gensim/
[12] https://code.google.com/archive/p/word2vec/

**GloVe.**   We use three sets of standard pretrained GloVe embeddings: 100D and 300D embeddings trained on Wikipedia and Gigaword (6B tokens), and 300D embeddings trained on Common Crawl (840B tokens).[13] We handle tokenization and embedding lookup identically to word2vec; for the Wikipedia/Gigaword embeddings, which are uncased, we lower case all tokens as well.

**FastText.**   We use four sets of pretrained FastText embeddings: two trained on Wikipedia and other news corpora, and two trained on Common Crawl (each with an original version and one trained on subword information).[14] We use the Python port of the FastText implementation to handle tokenization, embedding lookup, and OOV embedding computation.[15]

**ELMo.**   We use three pretrained models made available by AllenNLP: *small*, *original*, and *original (5.5B)*.[16] We use `spacy` to perform word tokenization, consistent with the `allennlp` library; we also use `allennlp` (0.7.2) to compute the ELMo embeddings. We average the embeddings over all three bidirectional LSTM layers.

**BERT.**   We use Hugging Face's `pytorch-transformers` (0.6.2) implementation and pretrained BERT base cased model.[17] To tokenize, we use the provided `BertTokenizer`, which handles WordPiece (subword) tokenization, and in general follow the library's recommendations for feature extraction.

For finetuning BERT on MultiNLI (matched subset), we generally use the default parameters provided in the library's `run_classifier.py` (batch size = 32, learning rate = 5e-5, etc.). We finetune for three epochs, and obtain 84.1% dev accuracy (reasonably consistent with the original work).

**GPT.**   We use the same Hugging Face library and associated pretrained model for GPT; we use their BPE tokenizer and `spacy` for subword and word tokenization respectively.

**InferSent.**   We use the authors' implementation of InferSent, as well as their pretrained V1 model based on GloVe.[18] (Unfortunately, the FastText-based V2 model was not available while performing the experiments in this paper; see issues #108 and #124 in the linked Github.) As per their README, we use the `nltk` tokenizer (3.2.5).

**Universal Sentence Encoder.**   We use pretrained models available on TensorFlow Hub for both the DAN and Transformer variants.[19] The modules handle text preprocessing on their own.

COMPUTATIONAL DETAILS

Experiments for ELMo, BERT, GPT, and the Transformer version of USE were run on a NVIDIA Titan XP GPU with CUDA 9.2. All other experiments were performed on CPUs.

# B   APPENDIX: ADDITIONAL RESULTS & PLOTS

## B.1   ANNOTATED N2O HEATMAP

The heatmap on the next page is a larger version of Fig. 3 that includes the N2O values ($k = 50, n = 100$, averaged over five runs).

---

[13]`https://nlp.stanford.edu/projects/glove/`

[14]`https://fasttext.cc/docs/en/english-vectors.html`

[15]`https://github.com/facebookresearch/fastText/tree/master/python`

[16]`https://allennlp.org/elmo`

[17]`https://github.com/huggingface/pytorch-transformers`

[18]`https://github.com/facebookresearch/InferSent`

[19]DAN: `https://tfhub.dev/google/universal-sentence-encoder/2`
Transformer: `https://tfhub.dev/google/universal-sentence-encoder-large/3`

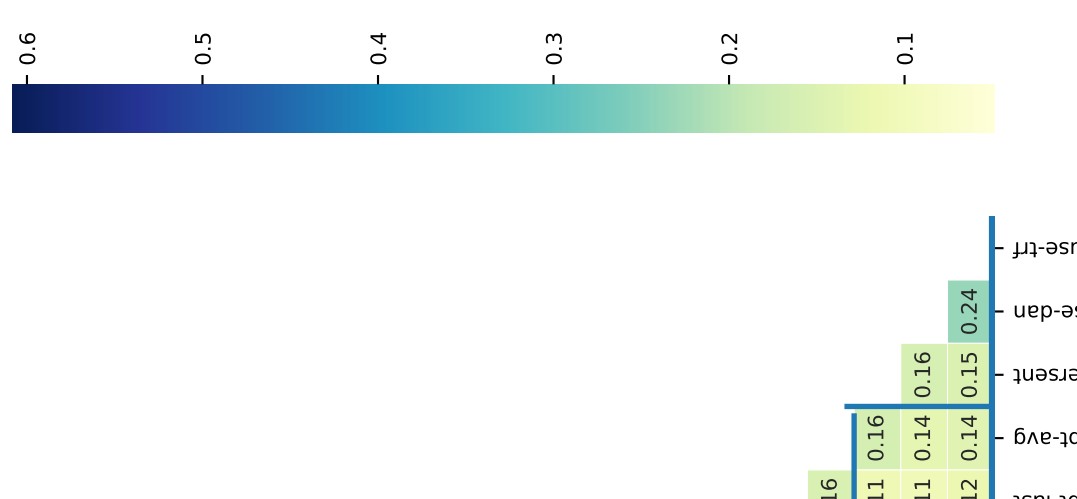

## B.2 TOKEN OVERLAP

In §5, we noted that the FastText subword variants had much lower N2O compared to other embedders (including analogues without subword information). Fig. 5 shows average token overlap between a query and its nearest neighbors, averaged over all queries. Unsurprisingly, `tfidf` has by far the highest overlap, and `fasttext-wiki-sub` and `fasttext-cc-sub` the lowest.

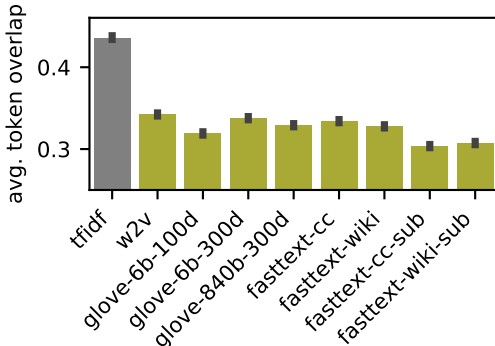

Figure 5: Average token overlap between a query and its nearest neighbors ($k = 50$), averaged over all queries. Error bars represent 95% confidence intervals.

## B.3 N2O RANGES PER EMBEDDER

In §5, we found that the BERT and GPT based embedders had low N2O with all other embedders, and averaging (rather than taking the `[CLS]` or last embedding) generally raised N2O. Fig. 6 shows boxplots of N2O values between each embedder and all other embedders.

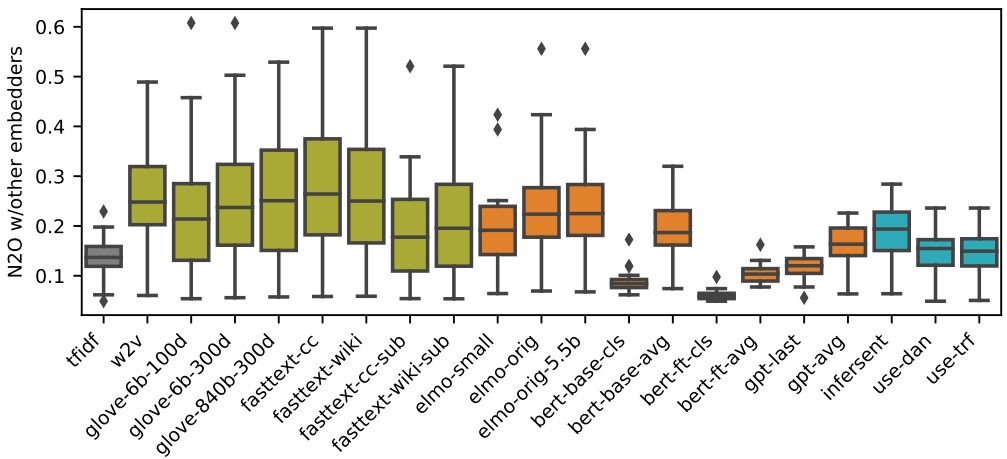

Figure 6: Comparison of N2O distribution between each embedder and all others.

### B.4 POPULAR & OUTLIER NEAREST NEIGHBORS

Table 3 shows additional outlier nearest neighbors from Table 2.

*Query*: Britain's biggest mortgage lender says that average house prices fell 3.6 percent in September, but analysts believe the market isn't that weak.

| Embedder | Rank | Sentence |
|---|---|---|
| all embedders | ≤ 5 | Average house prices in Britain fell 3.6 percent in September from a month earlier, the country's biggest mortgage lender said Thursday, although analysts believe the market isn't that weak. |
| bert-ft-cls | 2 | Japanese consumer prices fell for 13th straight month in March, though the GDP data suggests that deflationary pressures are starting to ease. |
| fasttext-cc-sub | 6 | It cautioned however that the economic situation abroad could still slow Sweden's recovery, and said the country's gross domestic product (GDP) would grow just 3.6 percent in 2011, down from its May estimate of 3.7 percent growth. |
| glove-840b-300d | 12 | Meanwhile, Australia's central bank left its key interest rate unchanged at 3.75 percent on Tuesday, surprising investors and analysts who had predicted the bank would continue raising the rate as the nation's economy rebounds. |

Table 3: Additional outlier near neighbors for the given query (top; same as Table 2). The first sentence is in the 5-nearest neighborhood for all embedders; the remaining sentences are highly-ranked by the given embedder and outside the 50-nearest neighborhood for all other embedders.

### B.5 QUERY-PARAPHRASE EXPERIMENT RESULTS

Table 4 shows results for the query-paraphrase experiment in §8: mean reciprocal rank (MRR), the number of queries for which its paraphrase was its nearest neighbor, and the number of queries for which the paraphrase was in its 5-nearest neighborhood.

| Embedder | MRR | # top | # top-5 |
|---|---|---|---|
| elmo-orig-5.5b | 0.910 | 67 | 70 |
| elmo-orig | 0.829 | 60 | 65 |
| infersent-v1 | 0.799 | 55 | 64 |
| w2v | 0.760 | 52 | 64 |
| use-trf | 0.759 | 54 | 60 |
| fasttext-cc | 0.756 | 52 | 62 |
| use-dan | 0.718 | 51 | 55 |
| bert-base-avg | 0.674 | 47 | 55 |
| glove-6b-300d | 0.673 | 48 | 52 |
| tfidf | 0.672 | 45 | 55 |
| fasttext-wiki | 0.662 | 45 | 54 |
| elmo-small | 0.638 | 44 | 51 |
| glove-840b-300d | 0.601 | 42 | 49 |
| gpt-avg | 0.600 | 41 | 50 |
| fasttext-wiki-sub | 0.552 | 37 | 47 |
| glove-6b-100d | 0.529 | 37 | 43 |
| fasttext-cc-sub | 0.515 | 35 | 41 |
| bert-ft-avg | 0.493 | 31 | 44 |
| bert-base-cls | 0.450 | 27 | 42 |
| gpt-last | 0.365 | 24 | 30 |
| bert-ft-cls | 0.302 | 19 | 27 |

Table 4: Results for the query-paraphrase experiment (§8), sorted by decreasing MRR. *# top* and *# top-5* are the number of queries for which the paraphrase was the nearest neighbor and in the 5-nearest neighborhood (max. 75), respectively.

## C  APPENDIX: APPROXIMATE NEAREST NEIGHBORS

As noted in §6, N2O computation is linear in the size of the corpus, and to have reasonable semantic overlap within a diverse set of sentences, the corpus should be large. The upfront cost of computing sentence embeddings across the corpus is unavoidable (and, for many applications, necessary anyways); our implementation of exact search is fast enough that repeated queries given precomputed embeddings is not a concern (see footnote 8).

However, we note that approximate nearest neighbor (ANN) methods are also a viable option, where computation of building an index of the corpus is front-loaded to ensure sub-linear search time. We recommend use of a small held-out set of queries to tune the ANN method parameters towards higher precision/recall (vs. speed).

All of the results in this paper were obtained using exact (linear) search. However, we also performed preliminary experiments using the NGT (neighborhood graph tree) library, which achieves good recall in high-dimensional settings (Iwasaki & Miyazaki, 2018; Aumüller et al., 2019).[20] We were able to obtain similar N2O-ranked results (query recall ∼0.96) relatively quickly: 0.25–5 s./query (depending on embedding dimension).

We note that, in related work, ANN is commonly used in retrieval settings; e.g., Sugawara et al. (2016) test multiple ANN methods for similar *word* embedding search, and Bhagavatula et al. (2018) use an ANN method to index documents for citation recommendation. We believe that approximate methods can be of use for scalable N2O computation as well.

---

[20]`https://github.com/yahoojapan/NGT`

