# OpenReview forum: "Situating Sentence Embedders with Nearest Neighbor Overlap"
_ICLR.cc/2020/Conference — Reject_

### Official Review · AnonReviewer3 · 2019-10-22
**Official Blind Review #3**

**Rating:** 3

**Review:**


The paper proposes N2O, a tool for probing the similarity among sentence embedders. Given two sentence embedders, N2O measures the amount of overlap of the k-nearest neighbor sets reported by the two embedders, averaged over a sample of probing queries. Cosine similarity is used as the similarity metric. The paper computes all-pair N2O scores for common sentence embedders and analyzes the results.

Overall, while the idea of probing similarity between embedders is interesting, the paper has the following weaknesses:

- The use of neighbor overlap to compare embedders has precedence in the context of word embeddings [https://www.aclweb.org/anthology/C16-1262/ | https://hal.archives-ouvertes.fr/hal-01806468/ | https://www.aclweb.org/anthology/Q18-1008/]. The methods in these works are mostly identical to N2O with some minor variations (e.g., using Jaccard distance).

- Using an existing method is justified if it provides new insights for the new setting. However, the results do not offer a lot of new insights. The fact that static embeddings, ELMo, and BERT give different neighbors is unsurprising. (The paper even admitted this.) Moreover, since sentence embedders are usually used as features when fine-tuned on downstream tasks, the importance of observations based on the pre-fine-tuned models is unclear. Contrast this with the work on word embedding similarity. Since some research areas (e.g., social science) directly use the similarity of word embeddings to conclude findings, the insights of how different word embeddings behave are more directly applicable.

- The paper does address some design choices, such as the number of neighbors and the number of probing queries, showing that different choices have little effects on the scores. However, the use of cosine similarity is problematic. As noted in the paper, some embedders such as ELMo were not trained on similarity objectives. BERT does have the next-sentence task, but the sparse attention-based architecture does not necessarily push similar sentences to have low cosine similarity.

Additional comments and questions:

- Instead of using the amount of overlap for a fixed k, it would be nice to have a metric that captures the whole distribution. For instance, maybe two embedders disagree on the first 20 neighbors, but end up retrieving the same set when considering 50 neighbors. The current overlap-based metric cannot capture such a phenomenon.

- Page 2: The method is technically not task-agnostic --- the task is sentence similarity with respect to a specific corpus.

- The paper tests variants of the same embedders by training on different corpora, with the conclusion that mismatched corpora give lower N2O scores. What is the N2O between two embedders of the same type trained on the same corpora but with different seeds?

**Experience Assessment:**

I have read many papers in this area.

**Review Assessment: Checking Correctness Of Derivations And Theory:**

I carefully checked the derivations and theory.

**Review Assessment: Checking Correctness Of Experiments:**

I carefully checked the experiments.

**Review Assessment: Thoroughness In Paper Reading:**

I read the paper thoroughly.

---

### Official Review · AnonReviewer1 · 2019-10-23
**Official Blind Review #1**

**Rating:** 1

**Review:**

The paper provides a method/metric for comparing sentence embedders, based on a nearest neighbor analysis. The method is straightforward: sample a sentence, embed the sentence and lots of sentences from a corpus, find the k nearest neighbors in the corpus to the sampled sentence, do the same for another embedder, calculate the overlap of the two sets of nearest neighbors.

The paper is commendably clear and easy to read.

The main problems I have with the paper are twofold: 1) it's not clear this is enough of a contribution for a top-tier ML conference; and 2) it's not clear what I do with the results.

I think the idea of analysing embedders in this way is potentially interesting, but it feels like the paper needs more. This analysis could be a great section in another paper (e.g. one which proposes another embedding method); or perhaps it could be extended in some way, so that the current content only takes up 1/2 the space, and then there's 1/2 the paper showing how useful this analysis is, for e.g. building better embedders for particular tasks.

The abstract starts by talking about how embedders have been evaluated using various benchmarks, and hints at the idea that this new comparative approach could be an alternative. But the new method can't really be used for evaluation: I don't come away from the paper knowing whether embedding method A is better than method B, only that A is more like B than C.

I think the problem with the paper as it stands is neatly summed up in the conclusion of the paper, which isn't a conclusion at all: it's just a mini-abstract. I'd like to know what readers should take away from the results, so that they can potentially build better embedders.

I've given the paper a 1. rating only because I really don't think it's ready for a full ICLR paper, not because I think the method is uninteresting or useless. On the contrary, with some more work and thought about how the analysis could be used, this could be a potentially useful tool.

**Experience Assessment:**

I have published in this field for several years.

**Review Assessment: Checking Correctness Of Derivations And Theory:**

I carefully checked the derivations and theory.

**Review Assessment: Checking Correctness Of Experiments:**

I carefully checked the experiments.

**Review Assessment: Thoroughness In Paper Reading:**

I read the paper thoroughly.

---

### Official Review · AnonReviewer4 · 2019-10-30
**Official Blind Review #4**

**Rating:** 1

**Review:**

The paper proposes a method to estimate the similarity of sentence embedders called N2O with the goal to better inform embedder choice in downstream applications. For two embedders A and B, N2O samples sentences called queries from a corpus, uses A and B to compute embeddings for each sentence, determines the k nearest neighbors (= other sentences from the corpus) for each sentence, and computes the overlap of the resulting sets of neighbors. Nearest neighbors are computed with Cosine similarity.

The paper should be rejected, mainly because the proposed method is not appropriate to inform embedder choices as claimed by the paper.

I doubt that the results from this paper can be used to make better embedder choices. The paper measures which embedders are similar in a specific way, but this does not tell us much about the performance in downstream tasks. Only based on the N2O measure, no informed decision can be made which embedder will perform well on a given task. Hence, the approach is not well motivated. Evaluation metrics such as the GLUE benchmark or SentEval provide much more information what we can expect from specific embedders. Hence, I think the results in the paper are only interesting to very few readers. Furthermore, the paper uses Cosine similarity to compute the similarity between embedders. However, embeddings are often not used directly, but used as input for a model (e.g. a neural network), which learns to predict something based on the embeddings. Hence, two embedders which are similar according to N2O may perform differently in downstream tasks.

To improve the paper, I recommend to test whether the N2O similarity correlates well with downstream performance. Concretely, the paper should clearly answer the question whether we can predict the downstream performance of an embedder A given its N2O similarity to an embedder B and the downstream performance of B. Furthermore, I recommend to also use other metrics to compute the distances between embeddings, and not only Cosine similarity. Th robustness and relevance of the findings is much higher if the findings are consistent across many different metrics. Instead of computing the overlap of the neighbors for different values of k, the paper could also use ranking measures to estimate the similarity of embedders. Furthermore, I think this paper fits much better to a conference focused on natural language processing.

Questions:
1. The paper claims that N2O can be used to inform embedder choices. Based on the main results in Figure 3, how can this help to decide which embedder to use for my task at hand?

**Experience Assessment:**

I have read many papers in this area.

**Review Assessment: Checking Correctness Of Derivations And Theory:**

N/A

**Review Assessment: Checking Correctness Of Experiments:**

I carefully checked the experiments.

**Review Assessment: Thoroughness In Paper Reading:**

I read the paper thoroughly.

---

### Author Response · Authors · 2019-11-14
**Author response**

Thank you all for your feedback! Some general comments:

* Correlation with downstream performance: Because embeddings are used in a wide variety of downstream tasks, we don’t expect (or want to impose) a single notion of “quality.” Taking a comparative approach like N2O helps us understand to what extent embedders are behaving similarly in the context of a target corpus; this is helpful, for example, in checking whether a new method is comparable to existing ones (high N2O).  More concretely, one could use N2O to decide which (small) subset out of the many different sentence embedders are worth comparing in a more expensive human evaluation or expensive search among methods.  N2O can identify when two embedders are functionally similar and therefore not worth both exploring.

* Use of cosine similarity: There are certainly other distance metrics we could try. We use cosine in this paper because it is the most frequently used in practice (it is the default in spacy and gensim, among other libraries).

    We agree that many embedders are not trained on similarity objectives, but nonetheless a common *expectation* of sentence embeddings is that distance is meaningful and semantic similarity <=> nearness in vector space. (See, for example, frequent questions on this matter in the BERT github issues.)  Direct comparison between sentence embeddings is useful in its own right for tasks like open-domain question answering, IE, and IR (e.g., for quickly identifying candidate spans), and has also seen increased use in computational social science analyses.

* Other ranking schemes: In general, we minimized the number of N2O hyperparameters where possible (e.g., weighting scheme by sentence rank or distance-based thresholding) to avoid premature assumptions about the embedding spaces & distortion of the results -- we view simplicity as a strength here.

---

> ### Comment · AnonReviewer4 · 2019-11-15
> **Does a high N2O scores really lead to similar downstream behavior?**
>
> Thanks a lot for your reply!
>
> I think a key point here is the statement: "N2O can identify when two embedders are functionally similar and therefore not worth both exploring.", which also summarizes the main claim/purpose of the paper.
>
> This statement is not substantiated by the experiments in the paper, because embeddings are usually used as input for a (potentially non-linear) layer/network. Hence, a high cosine similarity between two embedders A and B does not necessarily indicate that A and B will behave similarly in a downstream task. It would be great if you add an analysis to the paper to which extent a high cosine similarity (i.e. a high N2O score) indicates similar behavior in downstream tasks.

---

### Decision · Program_Chairs · 2019-12-19

**Decision:**

Reject

**Comment:**

This paper proposes to analyze the space of known sentence-to-vector functions by comparing the ways in which they induce nearest neighbor lists in a text corpus.

The primary results of the study are somewhat unclear, and the reviewers do not find the method to be novel enough—or sufficiently well motivated a priori—to warrant publication in spite of these results.